# Photocatalytic Inactivation of Viruses and Prions: Multilevel Approach with Other Disinfectants

**Takashi Onodera [1,2,\*], Katsuaki Sugiura [1], Makoto Haritani [1] , Tohru Suzuki [3] , Morikazu Imamura [4], Yoshifumi Iwamaru [5], Yasuhisa Ano [6] , Hiroyuki Nakayama [6,7] and Akikazu Sakudo [8]**

1   Laboratory of Environmental Science for Sustainable Development, Department of Veterinary Medical Sciences, Graduate School of Agricultural and Life Sciences, The University of Tokyo, 1-1-1 Yayoi, Bunkyo-ku, Tokyo 113-8657, Japan
2   Research Center for Food Safety, Graduate School of Agricultural and Life Sciences, The University of Tokyo, 1-1-1 Yayoi, Bunkyo-ku, Tokyo 113-8657, Japan
3   National Institute of Animal Health, Sapporo 052-0045, Japan
4   Faculty of Medicine, University of Miyazaki, Kiyotake-cho 889-1692, Japan
5   National Institute of Animal Health, Tsukuba 305-1102, Japan
6   Department of Veterinary Medical Sciences, Graduate School of Agricultural and Life Sciences, The University of Tokyo, 1-1-1 Yayoi, Bunkyo-ku, Tokyo 113-8657, Japan
7   Animal Medical Center, 3-60-7 Sendagaya, Shibuya-Ku, Tokyo 151-0051, Japan
8   School of Veterinary Medicine, Okayama University of Science, Imabari 794-8555, Japan
\*   Correspondence: atonode@g.ecc.u-tokyo.ac.jp

**Abstract:** Ag, Cu, Zn, Ti, and Au nanoparticles show enhanced photocatalytic properties. Efficient indoor disinfection strategies are imperative to manage the severe acute respiratory syndrome coronavirus 2 (SARS-CoV-2) pandemic. Virucidal agents, such as ethanol, sodium hypochlorite, 222-nm UV light, and electrolyzed water inactivate SARS-CoV-2 in indoor environments. Tungsten trioxide ($WO_3$) photocatalyst and visible light disinfect abiotic surfaces against SARS-CoV-2. The titanium dioxide ($TiO_2$)/UV system inactivates SARS-CoV-2 in aerosols and on deliberately contaminated $TiO_2$-coated glass slide surfaces in photocatalytic chambers, wherein 405-nm UV light treatment for 20 min sterilizes the environment and generates reactive oxygen species (ROS) that inactivate the virus by targeting S and envelope proteins and viral RNA. Mesoscopic calcium bicarbonate solution (CAC-717) inactivates pathogens, such as prions, influenza virus, SARS-CoV-2, and noroviruses, in fluids; it presumably acts similarly on human and animal skin. The molecular complexity of cementitious materials promotes the photocatalysis of microorganisms. In combination, the two methods can reduce the pathogen load in the environment. As photocatalysts and CAC-717 are potent disinfectants for prions, disinfectants against prionoids could be developed by combining photocatalysis, gas plasma methodology, and CAC-717 treatment, especially for surgical devices and instruments.

**Keywords:** influenza virus; prionoids; prions; SARS-CoV-2; $TiO_2$; tungsten trioxide

## 1. Introduction

Titanium dioxide ($TiO_2$), tungsten trioxide ($WO_3$), and graphene possess photocatalytic properties [1]. Ultraviolet (UV) rays induce the generation of hydroxy radicals, hydrogen peroxide, or reactive oxygen species (ROS). These biologically reactive species induce microbiocidal effects against a variety of organisms, such as bacteria, fungi, and viruses [2]. Recently, visible light has been reported to induce photocatalysis [3–5]. It can reduce the abundance of *Staphylococcus aureus*, *Enterococcus faecalis*, *Escherichia coli*, and bacteriophages on abiotic surfaces to levels below $10^{-2}$ to $10^{-6}$ cells per mL.

A complex salt consisting of silver iodide and benzalkonium chloride reportedly induced photocatalytic effects in commercially available painting solutions [6] and exhibited microbiocidal properties in the environment. The following three mechanisms are involved

in viral inactivation using TiO$_2$ complexes: structural disintegration of viruses, toxicity to metal ions among metal-containing photocatalysts, and chemical oxidation of ROS generated over the photocatalysts [7]. The total photocatalytic virucidal effect is much stronger than the original toxicity of metal ions, as shown by comparing the results in an environment without lighting. Photocatalyst-induced ultrastructural destruction in adenoviruses has been demonstrated using transmission electron microscopy [8].

Several nanomaterials such as those of Ag [9,10], Cu [11,12], Zn [13], Ti [14], and Au [15] have been used for developing photocatalysts. The Japanese government is continuing to pursue a more effective photocatalyst to manage the global pandemic. Some of these photocatalysts exhibit stronger virucidal effects than the original materials, which could help in developing new methods to prevent the transmission of severe acute respiratory syndrome coronavirus 2 (SARS-CoV-2) [16,17]. Although the detailed mechanisms underlying the function of these nanoparticles are not well-studied, the generation of ROS could be the main mechanism responsible for their virucidal effects. Recently, photocatalysts have been studied for their virucidal effects, such as those against SARS-CoV-2 [18], bovine coronavirus [19], influenza virus [20], tobacco mosaic viruses [21], phages [22], human adenovirus [8], human hepatitis B viruses [23], herpes simplex virus [13], as well as prions [24,25]. Viral inactivation using TiO$_2$ was dependent on the UVA light intensity and exposure time [20]. Inactivation of the virus in suspension was as efficient as that on a TiO$_2$-coated glass slide. Nakano et al. [20] also reported that viral proteins were primarily attacked during photocatalysis. After the destruction of the outer viral protein, the viral RNA was targeted.

This review primarily focuses on influenza viruses, SARS-CoV-2, and prions, owing to their involvement in major global pandemic incidents since 2000, including in the Japanese community. Recently, 222-nm UV rays have been shown to be effective against SARS-CoV-2; they do not exert any harmful effects on the skin eyes. This represents a major development in photocatalysis research. In this review, we briefly describe the prospects of using these UV rays for curtailing virus reproduction and transmission. Moreover, photocatalysts effectively reduce the infectivity of prions. Photocatalytic treatment is effective for destroying nucleic acids and proteins in vitro [26]. Photocatalytic treatment of liquid waste is a powerful tool for disinfection of the scrapie agent, and the described method could be adapted for use in surgical tool decontamination. Thus, we also discuss the application of photocatalysts for the treatment of diseases caused by prion-like amyloid or prionoids.

## 2. Decontamination of Influenza Viruses

The highly pathogenic avian influenza (HPAI) A (H5N1) virus, reported from avian species [27], is highly transmissible; and its infection is potentially fatal, particularly in domestic poultry. An Asian origin HPAI H5N1 virus has been responsible for high mortality in poultry and wild birds in Asia, the Middle East, Europe, and Africa since December 2000. To our knowledge, no H5 virus infections have been reported in humans in the United States or Japan to date [27]. HPAI viruses isolated from infected poultry cannot adapt to survive in human hosts [27]. However, sporadic infections due to other similar H5 viruses (e.g., H5N1 and H5N6) have been reported to afflict the human respiratory system and have high fatality rates [27]. Such infections have been reported in countries where personnel do not use appropriate personal protective equipment (PPE). PPE use is beneficial during direct physical contact with infected birds or virus-contaminated surfaces, i.e., when in the vicinity (within approximately 6 feet) of infected birds or visiting a live poultry market. Avian influenza infection in humans is unlikely when appropriately cooked poultry is consumed. However, direct or close (within 6 feet) contact with infected poultry or virus-contaminated equipment without using PPE may aggravate the risk of human infection [27]. Therefore, decontamination of infected birds or virus-contaminated surfaces is critical to public health.

Nakano et al. [20] expanded the photocatalysis effect of $TiO_2$ to the field of virology. In their study, they used A/PR8/H1N1 influenza virus and demonstrated photocatalytic inactivation of a viral suspension prepared in Dulbecco's modified eagle medium using $TiO_2$. The viral titer decreased by approximately 4 $\log_{10}$ in 8 h. Photocatalysis primarily destroys the ligand protein on the envelope, which could be involved in receptor binding on the cell surface [20] (Tables 1 and 2).

**Table 1.** Antiviral effects of photocatalysts.

| Photocatalyst | Agent | Pathogen Load | | Duration of Treatments | Reference |
|---|---|---|---|---|---|
| | | Photocatalysis-Untreated | Photocatalysis-Treated | | |
| Influenza | | | | | |
| $TiO_2$/Black Light (352 nm) | H1N1 | >$1.0 \times 10^{8.0}$ ($TCID_{50}$/mL) | undetectable | 8 h | [20] |
| Pt-$WO_3$ | H1N1 | $1.0 \times 10^{7.0}$ ($TCID_{50}$/mL) | <$1.0 \times 10^{1.5}$ ($TCID_{50}$/mL) | 6 h | [28] |
| SARS-CoV-2 | | | | | |
| $WO_3$ | JPN/TY/WK-521 | $5.98 \pm 0.38$ $\log_{10}$ ($TCID_{50}$/mL) | $3.05 \pm 0.25$ $\log_{10}$ ($TCID_{50}$/mL) | 6 h | [29] |
| $TiO_2$/LED | JPN/TY/WK-521 | $1.0 \times 10^5$ ($TCID_{50}$/mL) | undetectable | 2 h | [17] |
| Bovine coronavirus | | | | | |
| Peroxo titanium acid (70%) + peroxo-modified anatase | Hokkaido/9/03 | $4.4 \pm 0.3$ $\log_{10}$ ($TCID_{50}$/0.1 mL) | $1.6 \pm 0.1$ $\log_{10}$ ($TCID_{50}$/0.1 mL) | 4 h | [19] |

**Table 2.** Detailed conditions for each photocatalysis type.

| Photocatalysts | Agent | Reactor Volume | pH during Reaction | Temperature | Photocatalyst Retained Material | Light | Reference |
|---|---|---|---|---|---|---|---|
| Influenza | | | | | | | |
| $TiO_2$/Black Light | H1N1 | 100 μL | PBS | 25 °C | sprayed to glass with 1 g $TiO_2$ to 300 m² | UV (352 nm) | [20] |
| Pt-$WO_3$ | H1N1 | 100 μL | PBS | 25 °C | coated glass * | visible light (410–470 nm) | [28] |
| SARS-CoV-2 | | | | | | | |
| $WO_3$ | JPN/TY/WK-521 | 30 μL | 6.8 (MEM) + 2% FBS | 20 °C | 4 g/m² mixed with silica binder | visible light (>380 nm) | [29] |
| $TiO_2$/LED | JPN/TY/WK-521 | 1 mL | 6.8 (MEM) + 5% FBS | 20 °C | sprayed to glass fiber sheet ** | UV (405 nm LED) | [17] |
| Bovine coronavirus | | | | | | | |
| Peroxo titanium acid (70%) + peroxo-modified anatase | Hokkaido/9/03 | 150 μL | PBS | 25 °C | 0.2 mg/m² sprayed to projector film | visible light (>410 nm) | [19] |

FBS, fetal bovine serum; MEM, minimal essential medium; PBS, phosphate-buffered saline. * ILUMIO, produced by Sumitomo Chemical Co., Tokyo, Japan, ** produced by Kaltech Co. Ltd., Osaka, Japan. Detailed materials and methods are shown in each reference. The performance of the photocatalytic materials was considered effective when the antiviral activity of a photocatalyst with visible light irradiation ($V_L$) was ≥2.0 and the effect of visible light irradiation ($\Delta V$) was ≥0.3, based on the criteria of the Photocatalysis Industry Association of Japan [19,30]. $V_L$ of 2.0 and $\Delta V$ of 0.3 correspond to mean reductions of 1/100 and 1/2 of viruses, respectively, with visible light irradiation.

In 2009, a novel human influenza virus that originated from a swine influenza virus H1N1 (SIV) emerged in Mexico, and the infection was declared a pandemic by the World Health Organization on 11 June [28]. Precautionary regular hand washing was recommended because the primary routes for human transmission included aerosol and droplet infection. Abiotic surfaces, such as tables and walls, are additional sources of viral transmission through indirect contact. Virucidal treatment of these sources can reduce the virus transmission risk. Takehara et al. [28] used a platinum-loaded tungsten trioxide (Pt-$WO_3$)-based photocatalyst—ILUMIO (Sumitomo Chemical Co. Ltd., Tokyo, Japan) [28] to reduce the risk of viral transmission through abiotic surfaces. This Pt-$WO_3$-based photocatalyst was capable of absorbing visible light up to a wavelength of 470 nm and could efficiently break down organic material [28]; the avian isolated virus A/northern pintail/Miyagi/1472/18 (H1N1) was used in this study. The virus was exposed to a fluorescent lamp of $10^3$ lux without UV light for up to 120 min. The recovered virus was transferred to microtubes and cultured in Madin-Darby bovine kidney cells. The residual

viral load was below the detection limit i.e., >5.3 log inactivation within a 120-min exposure time [28] (Tables 1 and 2).

## 3. Decontamination of SARS-CoV-2

### 3.1. Decontamination of SARS-CoV-2 by Photocatalysts

The COVID-19 pandemic has brought into focus the identity and role of "reservoir hosts" (such as bats) [31], "novel hosts" (such as humans), and "intermediate hosts" (such as pangolins [32,33] and raccoon dogs), which act as a connecting link between the reservoir and novel host. Healthcare professionals prefer placing humans at the end of the chain of the COVID-19 advent; however, such a human-centric approach is sometimes insufficient to understand the true scenario. Preventive strategies against zoonotic diseases are limited. More rigorous and effective monitoring at the animal-human interface is presumably the easiest way to mitigate future pandemics [33]. However, wildlife harbors a large, diverse, and continuously evolving virus pool. Moreover, determining the infectivity of these viruses in human cells is time-consuming and costly. Similar to the application of social distancing to control COVID-19 transmission, efficient segregation of humans from wildlife and their associated viruses should be practiced to minimize viral mobility and host mortality. It is critical to establish some form of global "pandemic radar", which indicates sporadic zoonotic events with the potential for a full-blown disease outbreak. These systems should have ease of accessibility [34] and involve regular immunological surveillance to identify specific groups of viruses [35], such as coronaviruses.

Indoor disinfection strategies are important in controlling COVID-19 transmission, particularly in hospitals and zoos [36]. Reagents for indoor use with the highest virucidal activity against SARS-CoV-2 include mechanical ventilation [37], ethanol, sodium hypochlorite, and electrolyzed water [38]. Uema et al. used a tungsten trioxide ($WO_3$) visible light-responsive photocatalyst named RENECAT (Toshiba Materials, Kanagawa, Japan) against SARS-CoV-2 for a 6-h fluorescent light exposure at 20 °C (FL20SSW/18, Toshiba Lighting & Technology, Kanagawa, Japan) [29] and reported successful inactivation of SARS-CoV-2 on a glass slide surface coated with $WO_3$ (Tables 1 and 2).

Graphene derivatives have shown potential virucidal effects against SARS-CoV-2 [18]. To disinfect water using visible light, graphene oxide (GO) layers were integrated with $WO_3$ films (graphene-$WO_3$); this decreased the MS2 phage titer from $2 \times 10^6$ plaque-forming units (PFU)/mL to <5 PFU/mL [39]. In this method, light illumination below 400 nm was used for 180 min. Recently, the virucidal effects of graphene derivatives against SARS-CoV-2 have also been reported [40]. Graphene platforms with long alkyl groups (C11) have been used to understand the inhibitory effects of graphene against SARS-CoV-2. Strong inhibition of SARS-CoV-2 has been observed using graphene with polyglycerol sulfate coverage (G-PGS)-C11 using the plaque reduction assay of SARS-CoV-2 [18,40]. A review by Patial et al. indicated the role of nanomaterials, such as graphene, in inactivating SARS-CoV-2 and that it displayed dose-dependent inhibition efficacy [18]. However, this is preliminary research, and further studies are necessary.

Although the virucidal effects of $TiO_2$/UV as photocatalysts have been reported, the virucidal mechanisms of ROS remain unclear [41]. According to Liga et al. [42,43], non-enveloped viruses are more sensitive to the oxidizing effect of hydroxyl radicals than enveloped viruses. However, there is a lack of consensus on these observations. Nakano et al. [44] reported that non-enveloped viruses exhibit greater resistance to $TiO_2$/UV toxicity than enveloped viruses. They examined the susceptibility of feline calicivirus (FCV, non-enveloped virus) and influenza virus (IFV, enveloped virus) to the $TiO_2$/UV process. Compared to that of IFV, the titer of FCV required twice the time to decrease to a level below the detection limit. They concluded that the phospholipid membrane of the viral envelope transmits damage signals to the capsid protein much more rapidly than those in non-capsid viruses, such as FCV [44].

Recently, Matsuura et al. reported the inactivation of SARS-CoV-2 using the $TiO_2$/LED system on $TiO_2$-coated glass slides and aerosols in a photocatalytic chamber [17]. Pho-

tocatalytic chambers are used as electric air-cleaners, which operate using a 405-nm UV light for 20 min. This system induces ROS formation in the chamber, efficiently destroying virus S proteins, envelope proteins, and viral RNA. The SARS-CoV-2 suspension (1 mL; titer, $1.0 \times 10^5$ Median Tissue Culture Infectious Dose [$TCID_{50}$]/mL) was placed on a $TiO_2$-coated sheet; in the $TiO_2$ + light group, SARS-CoV-2 was exposed to light-emitting diodes for 120 min, and the viral titer was subsequently confirmed using the $TCID_{50}$ assay; the values were below the detection limit (<1.0 $TCID_{50}$/mL). However, the effectiveness of the photocatalyst must be evaluated in practical applications. Furthermore, the duration for which the photocatalyst retains its virucidal activity against SARS-CoV-2 needs to be determined through continuous viral exposure testing (Tables 1 and 2).

Additionally, Yoshizawa et al. reported that visible-light responsive photocatalyst, composed of 70% peroxo-titanium acid and 30% peroxo-modified anatase (Wako Filter Technology, Ibaraki, Japan), was effective in inactivating bovine coronaviruses (Figures 1 and 2) [19]. Six test films (three each of photocatalytically coated and uncoated films) were placed in Petri dishes. The viral suspensions were maintained on the films with or without fluorescent light during the test at 25 $\pm$ 3 °C at 1000 lux light exposure for 4 h. The photocatalyst reduced the viral load by 2.8 $\log_{10}$ $TCID_{50}$/0.1 mL [19] (Tables 1 and 2).

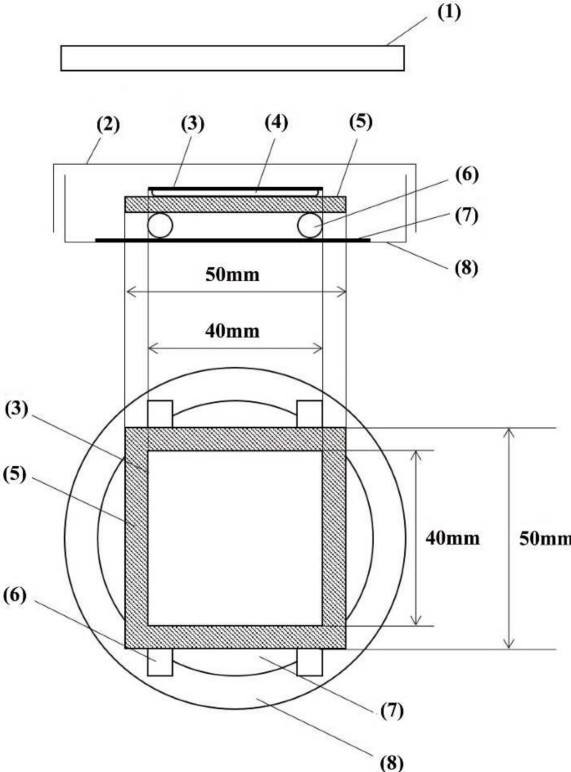

**Figure 1.** Schematic of the equipment used for the assessment of antiviral activity of photocatalytic materials using visible light irradiation. The equipment was prepared in accordance with the ISO 18071 criteria prescribed by the International Organization for Standardization. (1) Light source (fluorescent lamp), (2) Petri dish (upper), (3) cover film, (4) virus suspension, (5) test piece, (6) glass rod, (7) paper filter, and (8) Petri dish (lower). Reproduced from [19], published under an open access Creative Commons CC BY 4.0 license. UV experiments were conducted following Radiation Safety, https://www.mdpi.com/2076-0817/11/2/226 (accessed on 10 April 2020).

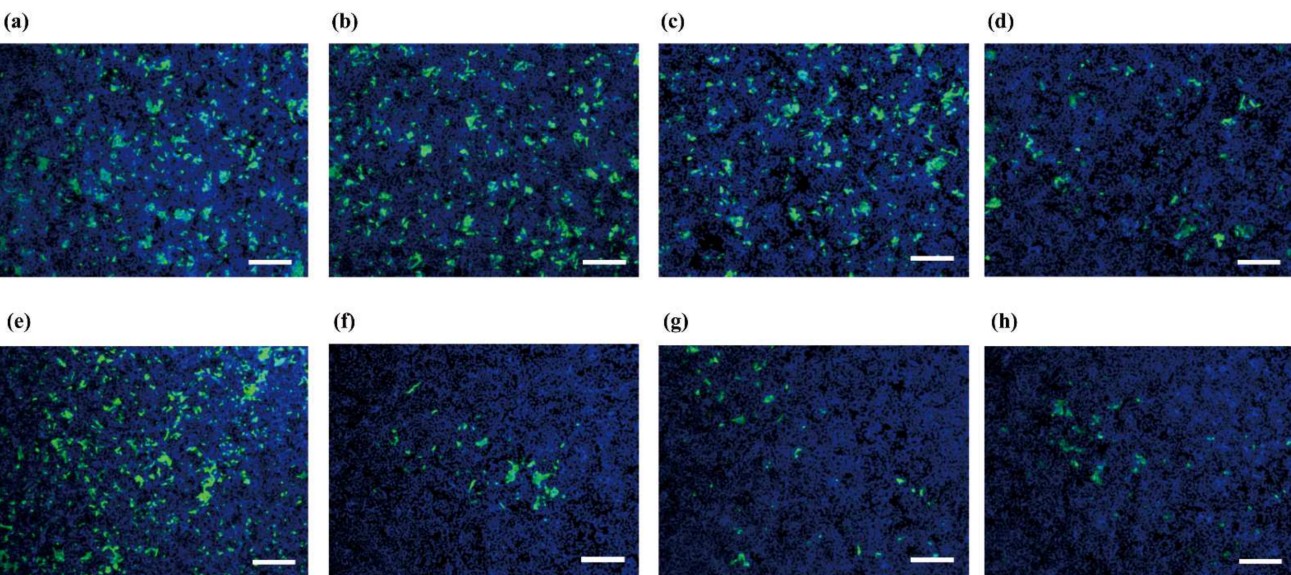

**Figure 2.** Images of the immunofluorescence assay (IFA) using viral solutions from photocatalytic (**a**–**c**) uncoated and (**d**–**h**) coated films were collected every hour (with or without irradiation; 1000 lux light for 4 h). After inoculation with the collected viral solutions for 22 h, the cells were stained with anti-BCoV rabbit serum, followed by incubation with goat fluorescein-isothiocyanate conjugate anti-rabbit immunoglobulin (Ig)G, IgA, and IgM (green) and finally with 4′,6-diamidino-2-phenylindole) (blue). (**a**) Photocatalytically uncoated films after 0 h without treatment with 1000 lux visible light; (**b**) photocatalytically uncoated films after 4 h without treatment with visible light of 1000 lux; (**c**) photocatalytically uncoated films after 4 h with treatment with visible light of 1000 lux; (**d**) photocatalytically coated films after 4 h without treatment with visible light of 1000 lux; (**e**–**h**) photocatalytic coated films after (**e**) 1 h, (**f**) 2 h, (**g**) 3 h, and (**h**) 4 h of treatment with 1000 lux visible light. Scale bar = 100 μm. Reproduced from [19], published under an open access Creative Commons CC BY 4.0 license.

TiO$_2$ is a useful material for photocatalytic degradation of pollutants in water [45]. Currently, TiO$_2$ is mainly immobilized in the form of films as self-cleaning coatings to avoid the loss of nanoparticles into water and allow long-term pollutant degradation. Moreover, hybridization with other materials can improve the physicochemical properties of the film. Particularly, silicon resin is a popular component used for nanoparticle-polymer complex coatings [46]. Double ligand phase spray deposition is widely used for TiO$_2$-silicon resin composition films [47] as it demonstrates excellent durability and photocatalytic activity under long-term hydrodynamic erosion. Excessive curing leads to the complete hardening of silicon resin and precludes the firm bonding of TiO$_2$ particles, resulting in excessive particle shedding and a decrease in photocatalytic efficiency [45,47]. Conditioning for an appropriately curved resin layer could efficiently fix the TiO$_2$ particles on the film on a glass substrate. This will allow the high photocatalytic efficiency to be preserved after long-term hydrodynamic scouring [45]. More studies are necessary to determine the ideal fabrication methods.

### 3.2. Decontamination of SARS-CoV-2 by 222-nm UV Light

222-nm UV light was recently reported to disinfect SARS-CoV-2-contaminated surfaces [48]. A low dose of 222-nm UV light exhibited very weak penetration of the skin and eyes [49]. The most common UV system for anti-microbe lamps involves 254-nm UV radiation C (UVC). Unfortunately, 254-nm UVC is harmful to the skin and eyes [50]. Far-UVC light (207–222 nm) exhibits an efficient germicidal effect and is less harmful to the skin and eyes [51]. Previous studies have reported that 222-nm UV light is effective in mitigating methicillin-resistant *Staphylococcus aureus* (MRSA) contamination in hospital-

use-only mobile phones [50]. Therefore, further studies should elucidate the use of 222-nm UV light to reduce the amount of SARS-CoV-2 in hospital aerosols.

## 4. Decontamination of Scrapie Prions

Prions are the causative agents of fatal neurodegenerative diseases such as transmissible spongiform encephalopathies (TSEs), including Creutzfeldt-Jakob disease (CJD) in humans, bovine spongiform encephalopathy in cattle, and scrapie in sheep and goats [52–54]. The only known component of a prion is the modified form of the cellular glycoprotein PrP$^C$, encoded by the *PRNP* gene [55], termed PrP$^{Sc}$. The central event in prion pathogenesis is the conformational conversion of PrP$^C$ into PrP$^{Sc}$ [52]. Extreme autoclaving conditions (134 °C, 18 min) are required to inactivate prions. Indeed, some prions retain transmissibility after dry-heating at 400 °C [56]. Prions are inactivated upon treatment with sodium dodecyl sulfate, sodium hydroxide, and sodium hypochlorite [56]; however, the use of these chemicals is impractical because of their corrosive properties. Paspaltsis et al. demonstrated that TiO$_2$-mediated photocatalytic oxidation significantly reduces the infectivity of scrapie prions [24], and Berberidou et al. reported the potential of homogenous photocatalysis using photo-Fenton reagent driven by UVA irradiation [25,57].

ROS induced during TiO$_2$/UVA application inactivate prions [24]. Paspaltsis et al. confirmed non-specific protein oxidation mediated by TiO$_2$ in the presence of UVA [24]. After treatment with TiO$_2$/H$_2$O$_2$, complete degradation of the brain protein (including prions) was observed. PrP$^C$ decomposition was observed after 30 min of TiO$_2$/UV treatment in the presence of 0.8% colloidal nano-TiO$_2$. PrP$^{Sc}$ disappeared after 60 min of treatment in the TiO$_2$/UV system. PrP$^{Sc}$ was more resistant to ROS than PrP$^C$ [24]. This phenomenon was attributed to the difference in protein secondary structure between PrP$^C$ and PrP$^{Sc}$ [58]. Thus, it was assumed that a different secondary structure implies a different physicochemical structure. However, many scientists believe that the secondary structure of PrP$^{Sc}$ is hypothetical [52]. Furthermore, it is impossible to study the PrP$^{Sc}$ secondary structure using nuclear magnetic resonance (NMR) because of the hydrophobic nature of PrP$^{Sc}$. PrP$^{Sc}$ was not detectable in Western blots after treatment with TiO$_2$, UVA, and H$_2$O$_2$. Another study showed that TiO$_2$ catalytic treatment is effective for destroying nucleic acids and proteins in vitro [26]. The photocatalytic treatment of liquid waste is a powerful tool for disinfection of the scrapie agent, and the described method could be adapted for the decontamination of surgical tools.

## 5. Future Prospects for Decontamination of Prionoids

### 5.1. Transmission of Amyloid A Amyloidosis and Bovine Aβ

Prionoid transmission causes various diseases in humans and animals [59]. Several protein aggregates linked to these disorders have been observed to undergo cycles of nucleation and fragmentation, as occurs in prions [59]. A study in co-isogenic *Prnp*-deficient mice for a thorough reassessment of the functions previously attributed to PrP$^C$ revealed novel functions of PrP$^C$ [34]. Contamination of surgical tools is the main route of transmission of prions [59]. Thus, TiO$_2$/UV-based disinfection of medical tools is necessary in human and veterinary medicine to reduce TSE transmission in the iatrogenic pathway.

Several prionoid-mediated disorders are characterized by systemic aggregate deposition in dialysis-related amyloidosis amyloid A in reactive amyloidosis [60]. This disease is mainly observed in the kidney glomeruli and includes aggregation of immunoglobulin light-chain amyloidosis, transthyretin in familial amyloid polyneuropathy, and β2-microglobulin. As a precursor protein, serum amyloid A (AA) is involved in the refractory systemic AA amyloidosis. This disease has also been reported in several domestic animals and birds. Murakami et al. observed AA amyloidosis transmission primarily in bovine and avian samples [60]. Nuclear formation and fibril extension were observed as the two steps of amyloid formation (Figure 3). Although there is species-specificity in AA transmission, interspecies transmissions are also observed occasionally. After AA is received from different species as a nucleus, the AA amyloid concentration increases via de-novo

proliferation. This proliferation involves host proteins and carbohydrates for pathogenesis. Inoculated AA or their precursors never proliferate in this system. AA proliferates de novo without using DNA or RNA systems [61]. Additionally, for transmission, large amounts of precursor AA proteins are needed. In this aspect, the de novo mechanism is different from prion replication. TSEs need a longer incubation period than AA, even under the increased levels of cellular prion protein content [61]; longer incubation periods are not necessary for bovine and avian amyloidosis.

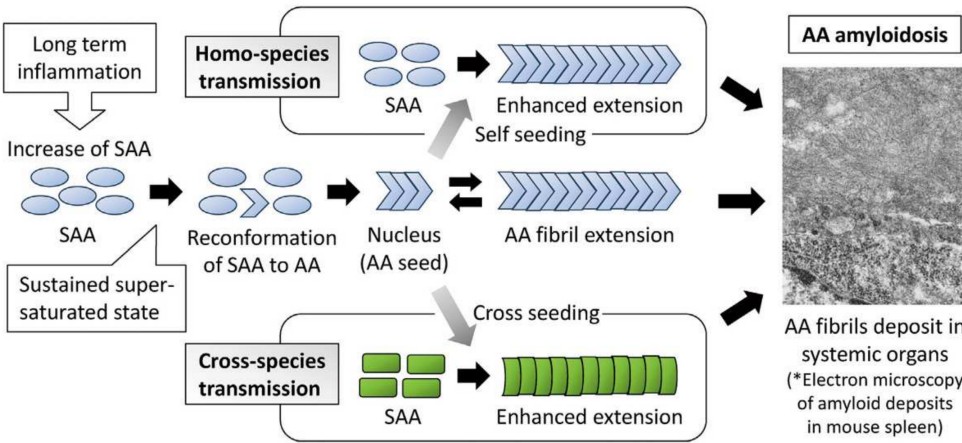

**Figure 3.** Diagram of amyloid fibril formation and transmission of AA amyloidosis among animals. Misfolded proteins are generated by reconformation, and they form the nucleus, which acts as a seed. The seeds interact with serum AA protein (SAA) and extend fibrils through the seeding-nucleation process. Fragmentation of AA fibrils sometimes generates new seeds. The fibrils or fibril fragments in feces or tissues are ingested by animals belonging to homo or cross-species, leading to homo-species transmission or cross-species transmission, respectively. Cited from Murakami et al. [60] with permission from Elsevier.

In-vitro studies on AA have shown that an equilibrium state is reached between the precursor protein and amyloid. Larger amounts of precursors are necessary to form amyloids. When the amount of precursor protein decreases, amyloid is refolded into precursor protein [62]. The quantity of amyloid fibrils can also be decreased by macrophage phagocytosis. In human patients with AA amyloidosis, when the SAA level is suppressed with medication, the amount of amyloid deposits in organs declines after a long period [63]. Since the $TiO_2$/UVA system decreases prion quantities, it may be useful in decreasing serum AA levels in these patients. There is a species-specificity similar to that observed for $PrP^{Sc}$ during cross-species transmission of AA amyloidosis.

Recently, many studies showed amyloid deposition in foods, such as beef and *foie gras* [64,65]. The amyloid enhancing factor (AEF) effect was seen in mice that were given *per os* with bovine AA (bAA) fibril [66]. There are concerns about human safety regarding AA amyloid transmission via amyloid-contaminated food [67]. However, it is unknown whether the transmission efficiency between different species is lower than that between the same species [68,69]. Iwaide et al. studied the oral transmission of bAA in 6-week-old mice and showed that fibrils consisting of bAA were derived from cows affected with hepatic AA amyloidosis [69]. None of the mice that were administered bAA developed amyloidosis; however, Peyer's patches were observed based on immunohisto-chemistry analysis. Notably, 1 day after the injection, only one out of three mice showed bAA in the mesenteric lymph nodes. The authors concluded that AA amyloid did not show bovine-mouse interspecies transmission [69]. Moreover, they did not detect amyloidosis and hypothesized that blood bAA level was not sufficiently high to induce AEF activity because oral adsorption of bAA was not sufficient. Oral adsorption of bAA may be more efficient in younger or suckling mice. However, more studies are required to validate this hypothesis [69].

Ano et al. studied the oral transmission of bovine amyloid β (Aβ) in 2-week-old suckling cows ($n$ = 3) and a 6-month-old cow ($n$ = 1) [70]. Bovine Aβ and enhanced green fluorescent protein (Aβ-EGFP) solution (10 mg/mL) were administered to the cows (300 mL); this process was repeated 12 and 15 h after the first administration. The animals were euthanized 3 h after the last administration. Aβ-EGFP was detected in the villous epithelium in the ileum of the 2-week-old cows. Based on the fluorescence detected via Aβ- EGFP, the authors determined that Aβ was distributed only in the cytoplasm of the intestinal epithelial cells (Figure 4). Immunocytochemistry using the anti-EGFP antibody confirmed these results [70]. This distribution was also observed with Aβ-EGFP administration in a 6-month-old cow; however, the distribution in the intestinal epithelial cells was less frequent than that in 2-week-old cows (Figure 5). Aβ-EGFP uptake through the intestinal epithelium could be a superior model for studying Aβ incorporation in food and animals. Further studies on AA and Aβ transmissions are necessary.

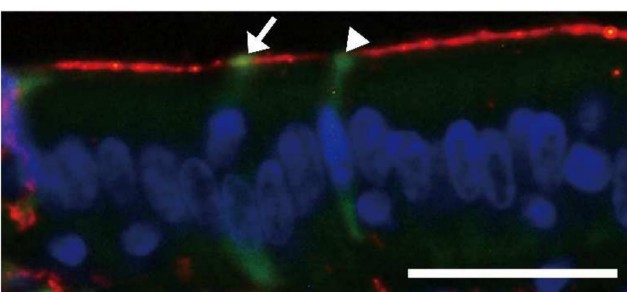

**Figure 4.** Identification of bovine Aβ and enhanced green fluorescent protein (Aβ-EGFP)-incorporating cells. Aβ-EGFP-incorporating cells were both villin positive (yellow on the cell surface, white arrow) and negative (green on the cell surface, white arrowhead). Microvilli of the epithelial cells were positive for villin (red) [70]. Nuclei were stained with 4′,6′-diamidino-2-phenylindole dihydrochloride (DAPI). Scale bar = 25 μm. Modified from [70] with permission from Wiley.

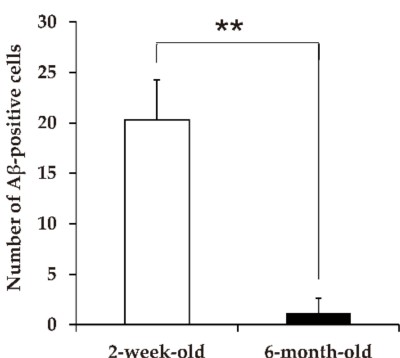

**Figure 5.** Age-dependent Aβ EGFP uptake by the intestinal epithelium. The number of ileal epithelial cells incorporating Aβ-EGFP was significantly higher in 2-week-old cows than in a 6-month-old cow. Results are expressed as the mean + standard deviation. Statistical differences were determined by a Student $t$-test. ** $p < 0.01$ [70]. Modified from [70] with permission from Wiley.

### 5.2. Induction of Aβ Deposition, Islet Amyloid Polypeptide Deposition, and Cerebral Aβ-Amyloid Angiopathy

Misfolded aggregates of islet amyloid polypeptides (IAPPs, amylin) are commonly observed in type 1 and 2 diabetes. IAPPs are reported as extracellular depositions from pancreatic islets [71]. IAPPs are produced in β-cells and secreted with insulin and move with blood circulation. They are misfolded in β-cells and develop the tendency to aggregate and mediate toxic oligomers [71]. An in-vitro study showed that the Islet IAPPs were misfolded and deposited in β-cells. This result indicated that IAPPs are prionoids [72]. IAPP deposition in pancreatic β-cells increases with the development of type 1 and 2 dia-

betes mellitus. Clinical manifestation involves hyperglycemia, impaired glucose tolerance, and a decrease in the number of pancreatic β-cells. Type 1 diabetes is accompanied by insulitis. IAPP accumulation plays an important role in the manifestation of type 1 and 2 diabetes, and reducing IAPP levels in blood circulation using photocatalysis can interfere with diabetes progression.

Iatrogenic induction of Aβ deposition is suspected in patients exposed to cadaveric pituitary-derived hormones, dura grafts, or surgical tools contaminated by Aβ [73]. Administration of human Alzheimer's disease (AD) brain homogenates containing Aβ aggregates to transgenic mice expressing human amyloid precursor protein induced cerebral β-amyloidosis and its pathology. However, Aβ-immunodepleted homogenates failed to induce lesions in mice. This suggests that amyloidosis induction is Aβ-dependent. Experiments on mice demonstrated that Aβ alone is sufficient for self-propagation in vitro, as in vitro-generated Aβ aggregates can induce amyloidosis [74]. Iatrogenic CJD (iCJD) has been demonstrated with dura matter grafting and is associated with Aβ accumulation in brain tissue [75]. This makes Aβ a strong candidate for prionoids [75]. A study revealed that patients with cadaver-derived human growth hormone (hGH)-related iCJD display tau-pathology that was linked to tau contaminations in respective hGH samples [76]. Confirmation of tau aggregates must be done using drug samples with or without tau. Thereafter, cadaveric hGH can be considered the possible source of tau transmissibility. Thus, further studies are required for validation.

Recently, eight human cases with iCJD, acquired via cadaver-derived hGH injection, were reported to have Aβ disease and pathology [77]. However, whether the hGH samples were the source of Aβ aggregation should be confirmed. Moreover, some patients showed tau-pathology; therefore, it must be clarified whether cadaveric hGH samples can induce tau-pathology. These patients may represent the first cases of iatrogenic Aβ transmission. Since then, there have been several reports of Aβ deposition following neurosurgical procedures involving dura matter grafting, as well as after cadaveric hGH inoculation in younger patients, from all over the world including Japan [78]. Although these authors did not suggest that AD is contagious [77], epidemiological studies have shown that it is transmissible [79,80]. Another report shows that it is important to evaluate the risk of iatrogenic transmission of cerebral Aβ-amyloid angiopathy (CAA) and potentially AD [81]. Currently, efficient disease-modifying drugs for AD are not available [81]. It is important to develop a method for disinfection against prionoid seeds from surgical tools. Application and development of photocatalysis-based methods should also be considered.

### 5.3. Aggregates of α-Synuclein

Aggregates of α-synuclein have been linked to neurodegenerative diseases, particularly Parkinson's disease (PD) [82]. Similarly, the brain homogenates of patients with multiple systemic atrophy (MSA) induced α-synuclein phosphorylation and aggregation in mice [82]. A typical characteristic of prions is maintaining infectivity after serial passages. Since α-synuclein showed transmissibility after passages, α-synuclein could be a prionoid. In a mouse model, aggregation of α-synuclein induced by the brain homogenates required the presence of a hemizygous human mutant α-synuclein gene [83]. PD mouse models exhibit a neurological disorder similar to human PD. These complicated results indicate that the α-synuclein aggregates are heterogenous in different diseases. They differ from original diseases not only in cell type (neuronal inclusion bodies in dementia with Lewy bodies and PD; glial inclusion in MSA) but also in their potential for further propagation in mice [83,84].

A 2016 study showed that spinal cord homogenates prepared from wild-type and α-synuclein-null mice could induce α-synuclein deposits in mice hemizygous for the mutated human α-synuclein transgene [85]. When the mutated human α-synuclein gene is present in mice, they develop α-synuclein pathology after intracerebral injection of α-synuclein-null mouse spinal cord homogenate [52]. Some unknown factor in the spinal cord was inducing PD in the central nervous systems. Another study on patients with PD [86] who were

transplanted with human embryonic neurons reported that although the original neurons were young at the time of transplantation, the grafts developed α-synuclein inclusions 10–24 years later. This shows that α-synuclein aggregates could spread from host to graft in a prion-like manner [87,88], which is a matter of concern in the field of neuroscience.

Currently, some proteins may be classified as prionoids, especially the aggregated forms of synuclein and amyloid A. Moreover, if they are shown to be infectious, they may be reclassified as true prions [59]. However, it is important to devise strategies for the disinfection of proteopathic seeds from surgical instruments as a precautionary measure. Previous studies on prion diseases have focused on protein misfolding disorders. Future studies on prions are likely to focus on understanding aggregation-induced toxicity.

## 6. Conclusions

Nakano et al. expanded the use of the photocatalysis exhibited by $TiO_2$ to the field of virology, showing its virucidal effects against the influenza virus [20]. However, many unknown factors are involved in the mechanisms behind the virucidal effect of $TiO_2$, except for ROS as their role has been elucidated. Because of the COVID-19 pandemic, several anti-viral compounds are being developed against SARS-CoV-2 using photocatalysts. The application of photocatalysis-based anti-viral compounds in indoor environments and public spaces is expected to decrease the viral load in aerosols. During the investigation of aerosols, 222-nm UV light was found to be effective against SARS-CoV-2 and did not exert any adverse effects on skin and eyes at least in mice and rats. Further evaluation of safety is needed to reduce the contamination of hospital or public buildings [48,49,89,90]. There are no conclusive studies on the use of 222-nm UV against prions. Since protein glycosylation in prions is sensitive to redox change [91], this redox system can control $PrP^C$ conversion to $PrP^{Sc}$. Thus, more studies are necessary to validate this mechanism. Moreover, as photocatalysis suppresses prion replication, effective conditions for surgical tools and dialysis machines are being explored to exclude the substances during the treatments.

Photocatalysts could be useful for the disinfection of abiotic materials indoors, such as walls, tables, and surfaces, using normal fluorescent light. For aerosol infections, photocatalyst filters could be useful to reduce the viral load in the air. Recently, a mesoscopic calcium bicarbonate solution (CAC-717) has been used to eliminate a variety of pathogens such as prions, influenza virus, SARS-CoV-2, and noroviruses [92–96] from fluids. CAC-717 can also act on the surfaces of human or animal bodies. A combination of the two methods could reduce the number of pathogens in the environment. Surface pH can be critical in promoting virucidal effects. SARS-CoV-2 is rapidly inactivated within 15 min at pH > 12 [94,96]. The inhibitory characteristics of cementitious materials also promote the photocatalytic activity of compounds that can inactivate microorganisms [97]. Furthermore, a gas plasma methodology is useful in disinfecting a variety of organisms [98,99]. Recently, there are increasing numbers of reports concerning prionoids in relation to such diseases as AD, PD, amyotrophic lateral sclerosis, and factors in metabolic diseases and cancer [59]. Since photocatalysts and CAC-717 are potent disinfectants for prions, disinfectants for prionoids may be developed by combining photocatalysis and CAC-717 treatment.

**Author Contributions:** T.O., M.H. and A.S. conceptualized the study. T.O. wrote the original draft. T.O., K.S., M.H., T.S., M.I., Y.I., Y.A., H.N. and A.S. wrote, reviewed, and edited the final draft of the manuscript. All authors have read and agreed to the published version of the manuscript.

**Funding:** Orix and Santa Mineral provided financial support for the preparation of this article.

**Institutional Review Board Statement:** Not applicable.

**Informed Consent Statement:** Not applicable.

**Data Availability Statement:** Not applicable.

**Conflicts of Interest:** The authors declare no conflict of interest.

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
