# Peer review of "Photocatalytic Inactivation of Viruses and Prions: Multilevel Approach with Other Disinfectants"

_2673-8007, doi:10.3390/applmicrobiol2040054_

Round 1

Reviewer 1 Report

This review paper, although it could be of great interest, requires some editing before it can be considered for publication in such an esteemed journal.

Please edit for grammar or use proofreading services that exist. I could not correct all the grammatical errors.

Here are some of the edits suggested: 

-Indicate the objectives of this review. What is the purpose/goal of this work? 

-What are the shortcomings of the disinfection strategies discussed in this review? What precautions should be taken? Please discuss. 

-line 33: Please change the sentence to "Virucidal reagents, such as ethanol, sodium hypochlorite, and electrolyzed water have been reported to inactivate severe acute respiratory syndrome coronavirus 2 (SARS-CoV- 34 2) in indoor environments".

- Using UV requires radiation safety. Indicate this in your text. Please read on radiation safety and cite https://www.mdpi.com/2076-0817/11/2/226 

-Line 36: Please change to "The TiO2/UV system inactivates aerosolized SARS-CoV-2 as well as deliberately contaminated TiO2 coated glass slide surface in photocatalytic chambers". 

-Line 38: Please change to "This induces reactive oxygen species (ROS) in the chamber, which effectively inactivates the virus by targeting virus S proteins, envelope proteins, and viral RNA.

- DELETE  line 43 "Recently there are increasing number of candidates of prionoids such as Alzheimer disease, Parkinson’s disease, and factors in metabolic diseases and cancer". It does not align with this work as we cannot disinfect human subjects under such conditions. 

-Line 54:  xenon lamp emits UV too. So saying that ultraviolet (UV) or xenon lamp will induce hydroxy radical, hydrogen peroxidase or reactive oxygen species (ROS) generation is confusing. please revise. 

-Line 82: Highly pathogenic avian influenza (HPAI) A (H5N1) virus is prevalent in avian species.  Provider citation/reference for this statement.

- Line 140: Please read and cite https://www.nature.com/articles/s41586-020-2169-0  for an example of pongolin  and see more and cite https://www.ncbi.nlm.nih.gov/pmc/articles/PMC6893680/ 

-Line 154: Explain that indoor environments rely on mechanical ventilation to minimize the spread of aerosolized pathogens such as SARS-CoV-2. mechanical ventilation can be complemented using UV. Read and cite https://www.frontiersin.org/articles/10.3389/fbuil.2021.725624/full 

-UV can be used in indoor environments and is effective. Not just chemicals. Please clarify in the paragraph starting line 154. 

- Line 242: ROS induced during TiO2/UVA application inactivates prions. Please provide a citation to this statement. 

-edit Line 490: The inhibitory characteristic of cementitious materials also promote the photocatalytic activity that can inactivate microorganisms. 

Author Response

To Reviewer#1

Response: We are grateful for your detailed comments and productive suggestions that have helped us improve the quality of this manuscript, especially your suggestions on the newly developed 222-nm UV system.

<This review paper, although it could be of great interest, requires some editing before it can be considered for publication in such an esteemed journal. Please edit for grammar or use proofreading services that exist. I could not correct all the grammatical errors.>

Response: The manuscript has undergone another round of editing and is now free from any grammatical errors.

Here are some of the edits suggested: 

<-Indicate the objectives of this review. What is the purpose/goal of this work? >

Response: The objective of this review has been included in L81-88.

In this review, we briefly describe the prospects of using these UV rays for curtailing virus reproduction and transmission. Moreover, photocatalysts effectively reduce the infectivity of prions. Photocatalytic treatment is effective for destroying nucleic acids and proteins in vitro [26]. Photocatalytic treatment of liquid waste is a powerful tool for disinfection of the scrapie agent, and the described method could be adapted for use in surgical tool decontamination. Thus, we also discuss the application of photocatalysts for the treatment of diseases caused by prion-like amyloid or prionoids.

<-What are the shortcomings of the disinfection strategies discussed in this review? What precautions should be taken? Please discuss. >

Response: The shortcomings and precautions have been included in the Conclusion section in L505-520. Also, I have added a comment about importance of cementitious materials.

Nakano et al. expanded the use of the photocatalysis exhibited by TiO2 to the field of virology, showing its virucidal effects against the influenza virus. However, many unknown factors are involved in the mechanisms behind the virucidal effect of TiO2, except for ROS as their role has been elucidated. Because of the COVID-19 pandemic, several anti-viral compounds are being developed against SARS-CoV-2 using photocatalysts. The application of photocatalysis-based anti-viral compounds in indoor environments and public spaces is expected to decrease the viral load in aerosols. During the investigation of aerosols, 222-nm UV light was found to be effective against SARS-CoV-2 and did not exert any adverse effects on human skin and eyes. However, there are no conclusive studies on the use of 222-nm UV against prions. Since protein glycosylation in prions is sensitive to redox change [90], this redox system can control PrPC conversion to PrPSc. Thus, more studies are necessary to validate this mechanism. Moreover, as photocatalysis suppresses prion replication, effective conditions for sur-gical tools and dialysis machines are being explored to exclude the substances during the treatments.

L529-531

The inhibitory characteristics of cementitious materials also promote the photocatalytic activity of compounds that can inactivate microorganisms

<-line 33: Please change the sentence to "Virucidal reagents, such as ethanol, sodium hypochlorite, and electrolyzed water have been reported to inactivate severe acute respiratory syndrome coronavirus 2 (SARS-CoV- 34 2) in indoor environments".>

Response: Thank you for this valuable suggestion. We have revised the sentence in L24–26.

Virucidal agents, such as ethanol, sodium hypochlorite, 222-nm UV light, and electrolyzed water inactivate SARS-CoV-2 in indoor environments.

<- Using UV requires radiation safety. Indicate this in your text. Please read on radiation safety and cite https://www.mdpi.com/2076-0817/11/2/226 >

Response: As suggested, we have included radiation safety in the “Figure 1” section at the end of manuscript in L252-253, and “Ethics approval and consent to participate” section  L544-546.

UV experiments were conducted following Radiation Safety, https://www.mdpi.com/2076-0817/11/2/226.

Ethics approval and consent to participate

UV experiments were conducted following Radiation Safety (available at: https://www.mdpi.com/2076-0817/11/2/226.)

<-Line 36: Please change to "The TiO2/UV system inactivates aerosolized SARS-CoV-2 as well as deliberately contaminated TiO2 coated glass slide surface in photocatalytic chambers". >

Response: The sentence in L27–28 has been revised as per your suggestion. We had to modify it a bit to meet the word count limit for the abstract.

The titanium dioxide (TiO2)/UV system inactivates SARS-CoV-2 in aerosols and on deliberately contaminated TiO2-coated glass slide surfaces in photocatalytic chambers,

<-Line 38: Please change to "This induces reactive oxygen species (ROS) in the chamber, which effectively inactivates the virus by targeting virus S proteins, envelope proteins, and viral RNA.>

Response: We have revised the sentence in L30-32 per your suggestion. We had to modify it a bit to meet the word count limit for the abstract.

405-nm UV light treatment for 20 min sterilizes the environment and generates reactive oxygen species, which inactivate the virus by targeting S and envelope proteins and viral RNA

<- DELETE  line 43 "Recently there are increasing number of candidates of prionoids such as Alzheimer disease, Parkinson’s disease, and factors in metabolic diseases and cancer". It does not align with this work as we cannot disinfect human subjects under such conditions.>

Response: Thank you for pointing this out. As suggested, we have deleted this sentence.

<-Line 54:  xenon lamp emits UV too. So saying that ultraviolet (UV) or xenon lamp will induce hydroxy radical, hydrogen peroxidase or reactive oxygen species (ROS) generation is confusing. please revise. >

Response: Thank you for this valuable suggestion. I have revised the sentence in L44-45.

Ultraviolet (UV) rays induce the generation of hydroxy radicals, hydrogen peroxidase, or reactive oxygen species (ROS).

<-Line 82: Highly pathogenic avian influenza (HPAI) A (H5N1) virus is prevalent in avian species.  Provider citation/reference for this statement>

Response: We have added a reference for this statement in L91.

The highly pathogenic avian influenza (HPAI) A (H5N1) virus, reported from avian species [27],

  1. CDC. HPAI a H5 Virus Background and Clinical Illness. Available online: https://www.cdc.gov/flu/avianflu/hapi-background-clinical-illness.htm (accessed on 8 May 2022).

<- Line 140: Please read and cite https://www.nature.com/articles/s41586-020-2169-0  for an example of pangolin  and see more and cite https://www.ncbi.nlm.nih.gov/pmc/articles/PMC6893680/ >

Response: As suggested, we have added references for the statement in L151-152.

“novel hosts” (such as humans), and “intermediate hosts” (such as pangolins [30,31] and raccoon dogs)

<-Line 154: Explain that indoor environments rely on mechanical ventilation to minimize the spread of aerosolized pathogens such as SARS-CoV-2. mechanical ventilation can be complemented using UV. Read and cite https://www.frontiersin.org/articles/10.3389/fbuil.2021.725624/full >

Response: Thank you for this valuable suggestion. We have added references for this statement in L169. Mechanical ventilation is important to reduce the amount of SARS-CoV-2 in the aerosol, especially when it is used with UVC. In addition, we have added another sentence in L511-515.

Reagents for indoor use with the highest virucidal activity against SARS-CoV-2 include mechanical ventilation [35],

During the investigation of aerosols, 222-nm UV light was found to be effective against SARS-CoV-2 and did not exert any adverse effects on skin and eyes at least in rats and mice. Further evaluation of safety is needed to reduce the contamination of hospital or public buildings.

<-UV can be used in indoor environments and is effective. Not just chemicals. Please clarify in the paragraph starting line 154. >

Response: Thank you for pointing the out. We have revised the text and added another paragraph in L237-246.

Photocatalytic chambers are used as electric air-cleaners, which operate using a 405-nm UV light for 20 min. This system induces ROS formation in the chamber, efficiently destroying virus S proteins, envelope proteins, and viral RNA. The SARS-CoV-2 suspension (1 mL; titer, 1.0 × 105 Median Tissue Culture Infectious Dose [TCID50]/mL) was placed on a TiO2-coated sheet; in the TiO2 + light group, SARS-CoV-2 was exposed to light-emitting diodes for 120 min, and the viral titer was subsequently confirmed

<- Line 242: ROS induced during TiO2/UVA application inactivates prions. Please provide a citation to this statement. >

Response: Thank you for pointing out this oversight. We have included a citation for the statement in L287.

ROS induced during TiO2/UVA application inactivate prions [24].

  1. Paspaltsis, I.; Kotta, K.; LagoudSakudo, A.; Lagoudaki, R.; Grigoriadis, N.; Poulios, I.; Sklaviadis, T. Titanium dioxide photocatalytic inactivation of prions. J. Gen. Virol. 2006, 87, 3125–3130. DOI:10.1099/vir.0.81746-.

<-edit Line 490: The inhibitory characteristic of cementitious materials also promote the photocatalytic activity that can inactivate microorganisms.>

Response: We have revised the sentence in L529-532. 

The inhibitory characteristics of cementitious materials also promote the photocatalytic activity of compounds that can inactivate microorganisms [96].

Reviewer 2 Report

This paper reviews performance in inactivating viruses and prions by using photocatalysts and disinfectants, and provide future prospects based on the findings of previous publications. Revisions are required for possible publication in Applied Microbiology. The comments for this manuscript are as follows:

1.    Line 99–104. Basic knowledge of photocatalyst is described here, and I recommend the authors to move this part to beginning of Introduction.

2.     Line 104. Why does the authors use “However” here?

3.     Line 270, 289–291. Check the font. Different fonts seem to be used.

4.   Chapter 5 is too long, and it is better for readers to provide some subsection (e.g., 5.1 …, 5.2 …, 5.3 …).

5. Fabrication of film-like materials immobilizing photocatalyst particles onto supporting material have been reported in many studies, and some studies have reported that the particles are released from the material owing to the degradation of the material. The authors should summarize the duration that can maintain performance in inactivation of pathogen (e.g., cumulative time of UV irradiation) in revised manuscript.

6. Pathogen is inactivated via photocatalysis, whereas dead or inactivated cell accumulated on photocatalyst particles may lead to decrease in effective surface area of the particles and decrease in the photocatalytic performance owing to scavenging the reactive oxygen species (competition between living and dead pathogen). Please provide the future perspectives considering the long-term operation of photocatalytic reactor for inactivating pathogens.

7.     Table 1. This table just contains the results of previous studies and seems not to be informative. The authors should summarize the experimental conditions as well to quantitatively compare the findings in each study. This table should include initial number of living pathogens, reactor volume, pH during reaction, temperature, solvent (if the inactivation experiment has conducted in water. e.g., pure water, artificial wastewater, or actual wastewater), dose or concentration of photocatalyst or amount of photocatalyst retained (coated) in the material, UV light intensity, and rate constant for inactivating target pathogen.

Author Response

[Reviewer#2]

<This paper reviews performance in inactivating viruses and prions by using photocatalysts and disinfectants, and provide future prospects based on the findings of previous publications. Revisions are required for possible publication in Applied Microbiology. The comments for this manuscript are as follows:>

<1.    Line 99–104. Basic knowledge of photocatalyst is described here, and I recommend the authors to move this part to beginning of Introduction.>

Response: As suggested, we have moved this information on photocatalyst to introduction (L72-75).

<2.     Line 104. Why does the authors use “However” here?>

Response: During revision this sentence was deleted as part of restructuring the sentence. With this deletion, the discrepancy has been avoided.

<3.     Line 270, 289–291. Check the font. Different fonts seem to be used.>

Response: Thank you for pointing this out. We have ensured that the font size is uniform in the revised manuscript.

<4.   Chapter 5 is too long, and it is better for readers to provide some subsection (e.g., 5.1 …, 5.2 …, 5.3 …).>

Response: We are grateful for your valuable suggestion, and accordingly, we have divided section 5 into subsections 5.1, 5.2, and 5.3.

5.1 Transmission of Amyloid A Amyloidosis and Bovine Aβ

5.2 Induction of Aβ Deposition, Islet Amyloid Polypeptide Deposition, and Cerebral Aβ-Amyloid Angiopathy

5.3 Aggregates of α-synuclein

<5. Fabrication of film-like materials immobilizing photocatalyst particles onto supporting material have been reported in many studies, and some studies have reported that the particles are released from the material owing to the degradation of the material.>

Response: We have added a new paragraph in L221-234 and text in L511-515 describing this information.

TiO2 is a useful material for photocatalytic degradation of pollutants in water [44]. Currently, TiO2 is mainly immobilized in the form of self-cleaning coatings to avoid the loss of nanoparticles into water and allow long-term pollutant degradation. Moreover, hybridization with other materials can improve the physicochemical properties of the film. Particularly, silicon resin is a popular component used for nanoparticle‒polymer complex coatings [45]. Double ligand phase spray deposition is widely used for TiO2-silicon resin composition films [46] as it demonstrates excellent durability and photocatalytic activity under long-term hydrodynamic erosion. Excessive curing leads to the complete hardening of silicon resin and precludes the firm bonding of TiO2 particles, resulting in excessive particle shedding and a decrease in photocatalytic efficiency. Conditioning for an appropriately curved resin layer could efficiently fix the TiO2 particles on the film on a glass substrate. This will allow the high photocatalytic efficiency to be preserved after long-term hydrodynamic scouring [44]. More studies are necessary to determine the ideal fabrication methods.

  1. Zhang, L.; Rao, L.; Wang, P.; Guo, X.; Wang, Y. Fabrication and photocatalytic performance evaluation of hydrodynamic erosion-resistant nano-TiO2-silicone resin composite films. Env. Sci. Pollution Res. 2019, 26, 4997–5007. DOI:10.1007/s11356-018-4054-z.
  2. Zuo, W.; Feng, D.; Song, A.; Gong, H.; Zhu, S. Effects of organic-inorganic hybrid coating on the color stability of denture base resins. J. Prosthet. Dent. 2016, 115, 103–108. DOI:10.1016/j.prosdent.2015.07.008.
  3. Xiang, H.; Ge, J.; Cheng, S.; Han, H.; Cui, S. Synthesis and characterization of titania/MQ silicon resin hybrid nanocomposite via sol-gel process. J. Sol-Gel Sci. Technol. 2011, 59, 635–639. DOI:10.1007/s10971-011-2538-0.

During the investigation of aerosols, 222-nm UV light was found to be effective against SARS-CoV-2 and did not exert any adverse effects on skin and eyes at least in rats and mice. Further evaluation of safety is needed to reduce the contamination of hospital or public buildings.

The performance of the photocatalytic materials was considered effective when the antiviral activity of a photocatalyst with visible light irradiation (VL) was ≥ 2.0 and the effect of visible light irradiation (△V) was ≥ 0.3, based on the criteria of the Photocatalysis Industry Association of Japan [19, 99]. VL of 2.0 and △V of 0.3 correspond to mean reductions of 1/100 and 1/2 of viruses, respectively, with visible light irradiation.

< The authors should summarize the duration that can maintain performance in inactivation of pathogen (e.g., cumulative time of UV irradiation) in revised manuscript.>

Response: Previous studies have not reported results based on cumulative irradiation; all studies have described continuous irradiation. However, for treatment with 222-nm UV, experiments on cumulative irradiation could be performed in the future. We have added this information in the revised text on L236-246 and references 47-50. Ref. 50 is very important, because it demonstrates the safety of 22-nm UV LED on mouse skin susceptible to regular UV in Ref. 44 irradiation.

3.2 Decontamination of SARS-CoV-2 by 222-nm UV Light

222-nm UV light was recently reported to disinfect SARS-CoV-2-contaminated surfaces. A low dose of 222-nm UV light exhibited very weak penetration of the skin and eyes [47, 48]. The most common UV system for anti-microbe lamps involves 254-nm UV radiation C (UVC). Unfortunately, 254-nm UVC is harmful to the skin and eyes [49]. Far-UVC light (207–222 nm) exhibits an efficient germicidal effect and is less harmful to the skin and eyes [50]. Previous studies have reported that 222-nm UV light is effective in mitigating methicillin-resistant Staphylococcus aureus (MRSA) contamination on hospital-use-only mobile phones [49]. Therefore, further studies should elucidate the use of 222-nm UV light to reduce the amount of SARS-CoV-2 in hospital aerosols.

  1. Kitagawa, H.; Nomura, T.; Nazmul, T.; Omori, K.; Shigemoto, N.; Sakaguchi, T.; Ohge, H. Effectiveness of 222-nm ultraviolet light on disinfecting SARS-CoV-2 surface contamination. Amer. J. Infect. Cont. 2021, 49, 299–301. DOI: 10.1016/j.ajic.2020.08.022.
  2. Mariita, R.M.; Davis, J.H.; Randive, R.V. Illuminating human noroviruses: A perspective on disinfection of water and surfaces using uvc, norovirus model organisms, and radiation safety considerations. Pathogens. 2022, 11, 226. DOI:10.3390/pathogens1020226.
  3. Kaiki, Y.; Kitagawa, H.; Hara, T.; Nomura, T.; Omori, K.; Shigemoto, N.; Takahashi, S.; Ohge, H. Methicillin-resistant Staphylococcus aureus Contamination of hospital-use-only mobile phones and efficacy of 222-nm ultraviolet disinfection. Amer. J. Infect. Control. 2021, 49, 800–803. DOI:10.1016/j-ajic.2020.11.011.
  4. Yamano, N.; Kunisada, M.; Kaidzu, S.; Sugihara, K.; Nishiaki‐Sawada, A.; Ohashi, H.; Yoshioka, A.; Igarashi, T.; Ohira, A.; Tanito, M.; et al. Long-term effects of 222-nm ultraviolet radiation C sterilizing lamps on mice susceptible to ultraviolet radiation. Photochem. Photobiol. 2020, 96, 853–862. DOI:10.1111/php.13269.

<6. Pathogen is inactivated via photocatalysis, whereas dead or inactivated cell accumulated on photocatalyst particles may lead to decrease in effective surface area of the particles and decrease in the photocatalytic performance owing to scavenging the reactive oxygen species (competition between living and dead pathogen). Please provide the future perspectives considering the long-term operation of photocatalytic reactor for inactivating pathogens.>

Response: Thank you for this suggestion. We have added future perspectives in L221–226. In Ref. 44, the authors show that fabrication is useful to resist long term hydrodynamic erosion. Silicon resin composite film exhibited better adhesion stability and photodynamic activity under hydrodynamic erosion conditions.

 See above.

TiO2 is a useful material for photocatalytic degradation of pollutants in water [44]

<7.     Table 1. This table just contains the results of previous studies and seems not to be informative. The authors should summarize the experimental conditions as well to quantitatively compare the findings in each study. This table should include initial number of living pathogens, reactor volume, pH during reaction, temperature, solvent (if the inactivation experiment has conducted in water. e.g., pure water, artificial wastewater, or actual wastewater), dose or concentration of photocatalyst or amount of photocatalyst retained (coated) in the material, UV light intensity, and rate constant for inactivating target pathogen.>

Response. As per your suggestion, we have revised Table 2. Pt-W3 and TiO2/LED are industrial products. The names of the companies are written at the bottom of the table. The data in the table have been obtained from information currently available in the public domain. Reference 19 was written by the authors of the present manuscript.

Please see the attached PDF file.

Reviewer 3 Report

1. The content of the manuscript was less.

2. The topic was SARS-CoV-2, but the references was not recently

Author Response

[Reviewer#3]

<1. The content of the manuscript was less.>

Response: Thank you for this valuable suggestion which has helped us improve the quality of this manuscript. We have added a new paragraph in L221-234 and a new section in L236-246.

See above.

TiO2 is a useful material for photocatalytic degradation of pollutants in water [44].

3.2 Decontamination of SARS-CoV-2 by 222-nm UV Light

Moreover, we have revised Table 2 in L476–503.

Also shown above

<2. The topic was SARS-CoV-2, but the references was not recently>

Response: We have added a new section in L236–246 See above.

Section 3.2, “Decontamination of SARS-CoV-2 by 222-nm UV Light”, contains updated information, although the information in this field is being updated every day. We hope that our changes will be considered acceptable.

Round 2

Reviewer 2 Report

Authors should consider the reviewer's comments carefully and have not provided adequate response to some comments. However, the manuscript has been revised and may be accepted for publication in Applied Microbiology.